# Typicality Assessment of Onions (*Allium cepa*) from Different Geographical Regions Based on the Volatile Signature and Chemometric Tools

**DOI:** 10.3390/foods9030375

**Published:** 2020-03-24

**Authors:** Sara Fernandes, André Gois, Fátima Mendes, Rosa Perestrelo, Sonia Medina, José S. Câmara

**Affiliations:** 1CQM—Centro de Química da Madeira, Universidade da Madeira, Campus da Penteada, 9020-105 Funchal, Portugal; saracsfernandes@hotmail.com (S.F.); andregois_97@hotmail.com (A.G.); 2032114@student.uma.pt (F.M.); rpm@staff.uma.pt (R.P.); 2Departamento de Química, Faculdade de Ciências Exatas e Engenharia, Universidade da Madeira, Campus da Penteada, 9020-105 Funchal, Portugal

**Keywords:** onions, volatile signature, solid-phase microextraction, gas chromatography-mass spectrometry, geographic origin

## Abstract

Onion (*Allium cepa* L.) is one of the main agricultural commodities produced and consumed around the world. In the present work, for the first time, the volatile signature of onions from different geographical regions of Madeira Island (Caniço, Santa Cruz, Ribeira Brava, and Porto Moniz) was tested with headspace solid-phase microextraction (HS-SPME/GC-qMS) and chemometric tools, showing that the volatile signature was affected by the geographical region of cultivation. Sulfur compounds, furanic compounds, and aldehydes are the most dominant chemical groups. Some of the identified volatile organic metabolites (VOMs) were detected only in onions cultivated in specific regions; 17 VOMs were only identified in onions cultivated at Caniço, eight in Porto Moniz, two in Santa Cruz, two in Ribeira Brava, while 12 VOMs are common to all samples from the four regions. Moreover, some VOMs belonging to sulfur compounds (dipropyl disulfide, 3-(acetylthio)-2-methylfuran), furanic compounds (dimethylmethoxyfuranone, ethyl furanone, acetyloxy-dimethylfuranone), and lactones (whiskey lactone isomer), could be applied as potential geographical markers of onions, providing a useful tool to authenticate onions by farming regions where the influence of latitude seems to be an important factor for yielding the chemical profile and may contribute to geographical protection of food and simultaneously benefiting both consumers and farmers.

## 1. Introduction

Onion (*Allium cepa* L.) is one of the main agricultural commodities produced and consumed around the world [1]. Because of globalization of the markets, consumers are increasingly interested to know the geographical origin of food along with assumed quality, health benefits, and high aroma values. Consequently, in many countries, the methods used to authenticate foodstuffs are rising both commercial and legal issues [2]. In Europe, the origin of food is one of the main authenticity parameters concerning food; hence, European Union (EU) legislation has introduced the regulatory framework for food products including Protected Designation of Origin (PDO), Protected Geographical Indication (PGI), and Traditional Specialties Guaranteed (TSG). Furthermore, in 2012, Optimal Quality Terms (OQT), “mountain product” and “product of island farming” were defined (1151/2012 EU Regulation) to advance the promotion and protection of the regional foods with special quality, to avoid food frauds and to become aware of, and adopt good agricultural practices [3]. In this context, the Rapid Alert System for Food and Feed (RASFF) of the European Commission was informed of more of the 600 notifications about adulterations or food frauds for the period between 2011 and 2016. Among them, there were many notifications concerning the geographical origin of foodstuff, which indicates the growing concern about this issue.

Chemical interactivity between plants and their environments is mediated through the synthesis of secondary metabolites. In this sense, certain processes may be the important sources of variation in the concentration of metabolites. These processes implicate long-term acclimation, environmental changes including both the biotic and abiotic components, geographical differences implicating multiple cultivars, or various ambient conditions of the growing region of the individuals of a species, especially when they have genetic homogeneity [4].

With regard to phytochemical composition, plants produce several classes of volatile organic metabolites (VOMs) that are a part of their metabolome and contribute to the sensory characteristic of foods, often determining their acceptability and aroma quality. Among the VOMs in onions, sulfur compounds are the most dominant volatiles in fresh onions as disulfides and trisulfides, in addition to aldehydes and terpene compounds in a lower extent [5]. Previously, it has been demonstrated that VOMs can be used to establish the label of the geographical origin of several matrices such as honey [6,7], olive oil [8], truffles [9,10], tomatoes [11], coffees [12], or wines [13]. Nevertheless, no previous reports exist concerning this issue in onion samples.

In the last years, several techniques as solid–liquid extraction (SLE), heat extraction, enzyme-assisted extraction, solid-phase extraction (SPE), and solid-phase microextraction (SPME) have been proposed to establish the volatile profile in food matrices [14]. SPME technique combined with gas chromatography-mass spectrometry (GC-MS) is known as the golden-standard methodology to determine the volatile composition of different matrices and, therefore, could be used as a suitable approach to assess aroma quality, authenticity, and typicality of onion samples [15].

For the authors’ knowledge, no studies have been carried out showing the volatile signature of onions from Spanish variety cultivated at different geographical origins, in order to evaluate the effect of growing region in its volatile pattern. This issue is worthy to be assessed since the volatile signature may be affected by agronomic and edaphoclimatic factors, representing the geographical origin one of the most significant determinants because of climate and soil particularities. In this context, the main purpose of this work was to establish the volatile signature of onions from Spanish variety cultivated at different geographical regions of Madeira Island (Caniço, Santa Cruz, Ribeira Brava, and Porto Moniz) as a powerful strategy to identify potential geographical authenticity markers and as a valuable way toward product protection and valorization.

## 2. Materials and Methods

### 2.1. Chemicals and Materials

All chemicals used in this trial were of analytical grade. Sodium chloride (NaCl, 99.5%) for adjusting the ionic strength was acquired from Panreac (Barcelona, Spain). Volatile organic metabolites standards used for identification were obtained from Sigma-Aldrich (Madrid, Spain), Acros Organics (Geel, Belgium), and Fluka (Buchs, Switzerland) with purity >98%. The individual stock solutions were prepared in ethanol (concentration of 500 mg/L) and stored at 4 °C. Helium (Air Liquide, Portugal) was used as the GC carrier gas. The glass vials, SPME fiber, and SPME holder for manual sampling were purchased from Supelco (Bellefonte, PA, USA). A C_8_ to C_20_ straight-chain *n*-alkanes (concentration of 40 mg/L in n-hexane) was purchased from Fluka (Buchs, Switzerland) in order to calculate the Kovats index (KI).

### 2.2. Onions Samples

Fresh onions (*Allium cepa* L.) from four different geographical regions of Madeira Island (Caniço, Santa Cruz, Ribeira Brava, and Porto Moniz), belonging to the same variety (Spanish variety) and harvested under the same traditional agricultural practices, were used in the analysis. These onions were purchased directly from producers. Table 1 shows the climate conditions and geographic coordinates of the regions of Madeira Island from which onion samples were harvested.

Five sets of onions (250 g/each) were obtained from four different areas in order to achieve a sample as representative of the region as possible, in as much as the reliability of the discrimination models will depend on the set of samples tested. All samples were transported in a refrigerated (2–5 °C) dark box to the laboratory and stored at −20 °C until analysis.

### 2.3. Extraction of Volatile Metabolites

For each HS-SPME assay, onions (*Allium cepa* L.) were cut into smaller pieces using a knife in a short time interval of 30 s and homogenized by dispersant Ultra Turrax DI 25 basic (IKAWERKE, Staufen, Germany). Then, 10 g onion sample was transferred to a 50 mL amber glass vial containing 1.5 g of NaCl and 5 mL of deionized water with a magnetic stir bar. The vial, tightly capped with a PTFE-faced silicone septum, was placed in a thermostatic block with constant magnetic stirring (750 rpm). The divinylbenzene/carboxen/polydimethylsiloxane (DVB/CAR/PDMS) fiber was exposed to the headspace for 30 min at 40 ± 1 °C. After SPME, the fiber was withdrawn into the holder needle, removed from the vial and immediately introduced into the GC injector port for 6 min at 250 °C for thermal desorption of the VOMs. These analytical conditions were chosen taking into account a previous work that reported the analysis obtained through absorption in the SPME fiber at the higher temperature (40 °C) compared to 20 °C isolated new compounds in onion samples [16]. All assays were performed in triplicate.

### 2.4. GC-qMS Conditions

An Agilent 6890N (Palo Alto, CA, USA) gas chromatography system combined with an Agilent 5975 quadrupole mass selective detector and splitless injector were used. Chromatographic separation was carried out by a BP-20 (30 m × 0.25 mm i.d. × 0.25 µm film thickness) fused silica capillary column provided by SGE (Darmstadt, Germany) with helium (Helium N60, Air Liquid, Portugal) as carrier gas at a flow rate of 1 mL/min (column-head pressure: 13 psi). An insert of 0.75 mm i.d. was used and the injector temperature was fixed at 250 °C. The GC oven temperature was held for 5 min at 45 °C, ramped to 150 °C at 2 °C/min (held for 10 min) and increased from 150 to 220 °C (held for 10 min) at 2 °C/min. The manifold, GC-qMS interface, and quadrupole temperatures were held at 180, 220, and 180 °C, respectively. In addition, MS detection was set in full scan, the ion energy used for the electron impact (EI) was 70 eV and the source temperature was 180 °C. The electron multiplier was set to the auto-tune procedure. The mass acquisition range, made in full scan mode, was 30–300 *m*/*z*, 1.9 spectra/s. Furthermore, VOMs identification was achieved in the following ways:(i)Comparison the GC retention times (RT) and mass spectra with those of the standard, when available;(ii)All mass spectra were also compared with the data system library (NIST, 2005 software, Mass Spectral Search Program v.2.0d; Nist 2005, Washington, DC, USA). Single VOM peak was considered as identified compound when its experimental spectrum corresponded with a score of over 80% that present in the library;(iii)Kovats’s retention index (KI) values were determined according to the van den Dool and Kratz equation [17]. For the determination of the KI, a C_8_–C_20_
*n*-alkanes series were used and the values were compared, when available, with values reported in the scientific literature for similar columns [18,19,20].

### 2.5. Statistical Analysis

The differentiation of onion samples from the different geographic origins based on VOMs profile was performed by principal component analysis (PCA) applied to the auto scaled areas of VOMs using SPSS statistical package version 25 (Chicago, IL, USA). Auto-scaling is a process of prior data processing, which allows the variables of different scales comparable. Each variable is automatically scaled separately by subtracting its average value and dividing by its standard deviation. In addition, the hierarchical clustering analysis (HCA) was processed employing the ward’s minimum variance algorithm method and the distance was determined using squared Euclidean distances by MetaboAnalyst 3.0. Also, HCA results are shown as a dendrogram. The objective of these approaches was to extract the main sources of variability and, therefore, to contribute in the characterization of the dataset.

## 3. Results

### 3.1. Characterization of Volatile Signature in Onions

The volatile signature of Spanish onions from four distinct areas of Madeira Island (Caniço, Porto Moniz, Ribeira Brava, and Santa Cruz) was investigated using headspace solid-phase microextraction (HS-SPME/GC-qMS). A total of 86 VOMs, 75 of them belong to ten different chemical groups (36 sulfur compounds, 11 furanic compounds, nine aldehydes, two terpenes, five esters, four lactones, three ketones, two acids, two alkanes, one alcohol) were identified (Table 2). In addition, 11 unknown volatiles were also detected (Table 2).

Table 2 shows all the detected VOMs, modes of identification, retention times, GC peak area (×10^5^), and relative standard deviation (RSD) (*n* = 3). The Kovats’s index (KIs) for each VOM was calculated under identical chromatographic conditions as the samples and then compared with the scientific literature to ensure the unequivocal identification of the VOMs. A representative typical GC-qMS chromatogram of the volatile signature of Spanish onions cultivated at Caniço, Santa Cruz, Ribeira Brava, and Porto Moniz is shown in Figure 1.

The results revealed that the volatile signature of Spanish onions from Porto Moniz was much richer (highest total GC peak area) compared to the volatile pattern of Spanish onions cultivated in the other investigated regions. A total of 58 VOMs were identified in Spanish onions cultivated in Caniço, whereas in Spanish onions from Porto Moniz, Santa Cruz, and Ribeira Brava were identified 56, 40, and 28 VOMs, respectively. Among all detected VOMs, 17 were only found in Spanish onions cultivated in Caniço, eight in Porto Moniz, two in Santa Cruz, two in Ribeira Brava, while 12 VOMs were common to all samples from four target regions (none of them belonging to sulfur compounds) (Table 2).

It was well-known that sulfur compounds are the main volatiles in fresh onions. In this regard, dipropyl disulfide was the major sulfur-identified compound in onions of Spanish variety cultivated in Ribeira Brava and Porto Moniz, followed by phenylethylthiol. Within the aldehydes, the highest VOM was 2-methyl-2-pentenal isomer (onions from the Porto Moniz region showed higher GC peak area compared to other ones). Related to furanic compounds, it can be mentioned that dimethylmethoxyfuranone and ethyl furanone presented the highest GC peak area (both in Porto Moniz samples), conditioning the sensory attributes of the Spanish onions. Based on its chemical structure, the identified VOMs were grouped into different chemical families, as is shown in Figure 2.

This study revealed that sulfur compounds, aldehydes, furanic compounds, and lactones were the main chemical groups of volatile composition in onions from Spanish variety from all investigated geographical regions. Nevertheless, it should be mentioned that there was an increase of specific chemical groups content (in terms of total GC peak area), highlighting a rise of furanic compounds, lactones, and alcohols in onions from Spanish variety cultivated at Porto Moniz and ketones and fatty acids in Caniço Spanish onions. This increasing trend is determined by the high levels of some individual VOMs (Table 2). Dimethylmethoxyfurarone and ethyl furanone were only detected in these two regions, the levels of onions from Porto Moniz being significantly higher (*p*-value < 0.05) than the onions grown in the other regions under study. Regarding lactones, whiskey lactone isomer (72) and γ-nonalactone (80) were detected at the highest levels in Spanish onions from Porto Moniz. In addition, undecanol (58) was only identified in Spanish onions from Porto Moniz suggesting that it could be a good candidate as a possible geographical authenticity marker.

### 3.2. Discrimination of Onions According to Geographical Origin

By application of PCA to the Spanish onions dataset, the first two principal components (PCs) accounted for 92% of the total variability. A score plot for the onions of Spanish variety according to the geographical region of cultivation is shown in Figure 3A.

The first principal component (PC1) explains 60% of the variability in the initial dataset and the second (PC2) explains 32%, allowing us to observe four clusters clearly defined: a group constituted by onions from Porto Moniz, another group by Ribeira Brava onion samples, and two more groups constituted by onions from Caniço and Santa Cruz, which are located close to each other (Figure 3A). This fact might indicate that the identified volatile metabolites seem strongly influenced by specific factors of each geographical region namely agronomic and edaphoclimatic conditions.

The PCA loading plot (Figure 3B) displays the contribution of the major VOMs for the observed discrimination of onions from Spanish variety. The variables with values near to zero had similar levels in all onion samples, regardless of location. Nevertheless, the PCA showed good separation of the onion samples according to geographical origin and several VOMs could be labelled as authenticity markers, which could potentially be used for the easy differentiation of the origin of the investigated onions. In this sense, Porto Moniz samples were projected in PC1 negative and PC2 positive axes being characterized by dimethylmethoxyfuranone (50), ethyl furanone (65), and whiskey lactone isomer (72), whereas 3-(acetylthio)-2-methylfuran (59) are strongly associated to onions cultivated at Caniço. The cluster of Ribeira Brava onions is placed in PC1 and PC2 positive axes being mainly characterized by dipropyl disulfide (26). These specific VOMs, responsible for the onions discrimination, belong to sulfur compounds, furanic compounds, and lactones.

The phytochemical composition is affected by factors such as genetic diversity, agricultural practices, and pedo-climatic factors based on a combination of land, soil and climate aspects (*terroir*). To assess the differences in VOMs among onions from Spanish variety cultivated in different regions of Madeira Island, hierarchical cluster analysis (HCA) was performed. A dendrogram based on Euclidean distance and Ward’s linkage is shown in Figure 3C. Therefore, it was possible to classify onions from Spanish variety based on its volatile signature. Thus, onions from Spanish variety from Caniço and Santa Cruz were clustered very close suggesting the similarity in volatile signature between onions cultivated in these regions. In addition, as we can see in Table 1 that the latitude and longitude coordinates of these geographical regions indicate that they are geographically close regions. Thus, the geographical origin, and particularly the latitude of the location seems to be an important factor for yielding the chemical profile of onions, showing a greater effect than altitude. In addition, the clustering result in the form of a heatmap for onions from Spanish variety grown in different regions of Madeira Island is shown in Figure 4.

## 4. Discussion

The organoleptic and the nutritional properties of onions have caused a great level of interest in the food field [21] and innovative metabolomics approaches have been developed for the detection of changes in onion metabolome during cold storage [22]. In other cases, segregation of different varieties based on their metabolite profile or due to several growing regions have been investigated [23,24]. However, it should be taken into account that the production of regional and traditional products depends on natural climatic factors and the determination of origin might be very difficult, but not impossible [25]. In this context, the current study assessed the typicality of onions from different geographical regions of Madeira Island based on the volatile signature, which shall serve to identify the optimal region suitable for onion cultivation. Indeed, the volatile profile of the onions found in the current study agrees with previously published data in fresh onion samples where most VOMs were sulfur-derived compounds [26,27,28]. Therefore, the sensory quality of onions appeared not to be generated from the individual VOMs but instead, emerged from the complex mixture of them, which could exhibit synergistic and masking effects as previously reported [5]. Particularly, furanic compounds that presented the highest levels in Porto Moniz samples constitute a major chemical group formed during the Maillard reactions being present not only in thermally treated foods but also in fresh vegetables, although in smaller proportion [29], providing aroma attributes such as dried and toasted fruits [30]. These compounds appear to be very important in the typicality of onions from this specific geographical region.

Nowadays, to prevent the food frauds in the global marketplace, it is necessary to apply analytical approaches such as metabolomics (including as volatilomic pattern) in conjunction with chemometrics analysis for the discrimination of foodstuffs according to geographical origin, an issue very relevant in the food authenticity and traceability. In this context, onions origin recognition has received great attention. In fact, the correlation between the geographic origin of onion and their trace element concentration has been investigated [31]. The models established by discriminant analysis (DA) or PCA based on multielement data are generally employed to differentiate the geographic origin of samples. In this regard, PCA is commonly used to reduce the data dimensionality while preserving, as far as possible, the information present in the original data, emphasizing the variation and drawing a strong pattern in a dataset matrix [2]. These variations can allow the successful discrimination of onions from different regions. In a recent study, considerable differences were observed in aromatic and amino acid profiles between onion samples from different geographic origins (Korean and Chinese) by metabolomics using NMR spectra [32]. The metabolomics approach has also been successfully implemented for the discrimination of several foodstuffs based on its origin. A previous study conducted by Gan et al. [33] investigated the geographical origin of monovarietal clarified apple juices, reported that carbonyl compounds and alcohols are responsible for its discrimination according to country production [33]. Furthermore, qualitative differences in sulfur compounds and terpenes between truffles originating from different Italian regions were found, suggesting that it might be possible to use the VOMs profile for the establishment of the origin of unknown samples [9,10]. In the same context, differences in levels of lactones, fatty acids, alcohols, and esters enabled the discrimination between wines produced in Madeira Island from those produced in Azores and Canary Islands [13].

On the other hand, latitude has a strong impact on the local climate environment and the latitude gradients of cultivation areas could influence on the chemical profile of agro-based foods, also lower and higher latitudes samples seemed to be clearly distinguishable [34]. This fact has also been previously reported by Romero and colleagues [35], in olive oil samples from different geographical origins, where they observed that the chemical composition of olive oils responsible for aroma (VOMs and phenols) varied as a function of temperature and latitude. They concluded that latitude between the orchards raised the significance of the geographical origin on the olive oil metabolome [35]. In the same sense, as stated by Kim and colleagues, in cabbage cultivars grown in different geographical locals, the factors responsible for its discrimination might be associated with environmental factors, like climate, soil type, and fertilization, among others. These authors enlightened that the discrimination of the samples depended on the sample origin rather than the cultivar [36].

Furthermore, and in connection with the biosynthesis pathways implicated in the formation of VOMs, it might be directly affected by the different expression of the enzyme pool. The availability of these enzymes, such as LOX (lipoxygenase), HPL (hydroperoxide lyase) or ADH (alcohol dehydrogenase), among others, could be dependent on the climatic conditions and therefore, so and we have mentioned above, the VOMs profile could be employed as “potential markers” for the identification of geographical scope [11]. A study carried out by Kumar and colleagues reported that LOX activity in soybean seeds was influenced significantly by the growing locations [37]. A variation in the levels of VOMs (mainly C_6_ aldehydes and alcohols) from virgin olive oil samples from different regions suggests that environmental growth conditions may influence the activity of ADH, which could explain the differences in the levels of specific VOMs [35].

## 5. Conclusions

To the best of our knowledge, this study represents the first characterization of the volatile signature of onions from Spanish variety cultivated in the different geographical areas of Madeira Island. HS-SPME/GC-qMS methodology combined with multivariate statistical analysis revealed a useful approach to discriminate the geographical origin of Spanish onions crops. The volatile composition of the Spanish onions is affected by the geographic region of harvest, both qualitative and quantitative expression. VOMs belonging to sulfur compounds (dipropyl disulfide (26), 3-(acetylthio)-2-methylfuran (59)), furanic compounds (dimethylmethoxyfuranone (50), ethyl furanone (65) and acetyloxy-dimethylfuranone (74)), and lactones (whiskey lactone isomer (72)), could be applied as authenticity markers of Spanish onions cultivated in Madeira Island, contributing within regulatory framework to assessment of the samples with claims such as “protected geographical indication” or “product of island farming.” However, authenticity markers validation in the food science field is a slow process and further investigations in this sense (with multi-year assay) are required.

## Figures and Tables

**Figure 1 foods-09-00375-f001:**
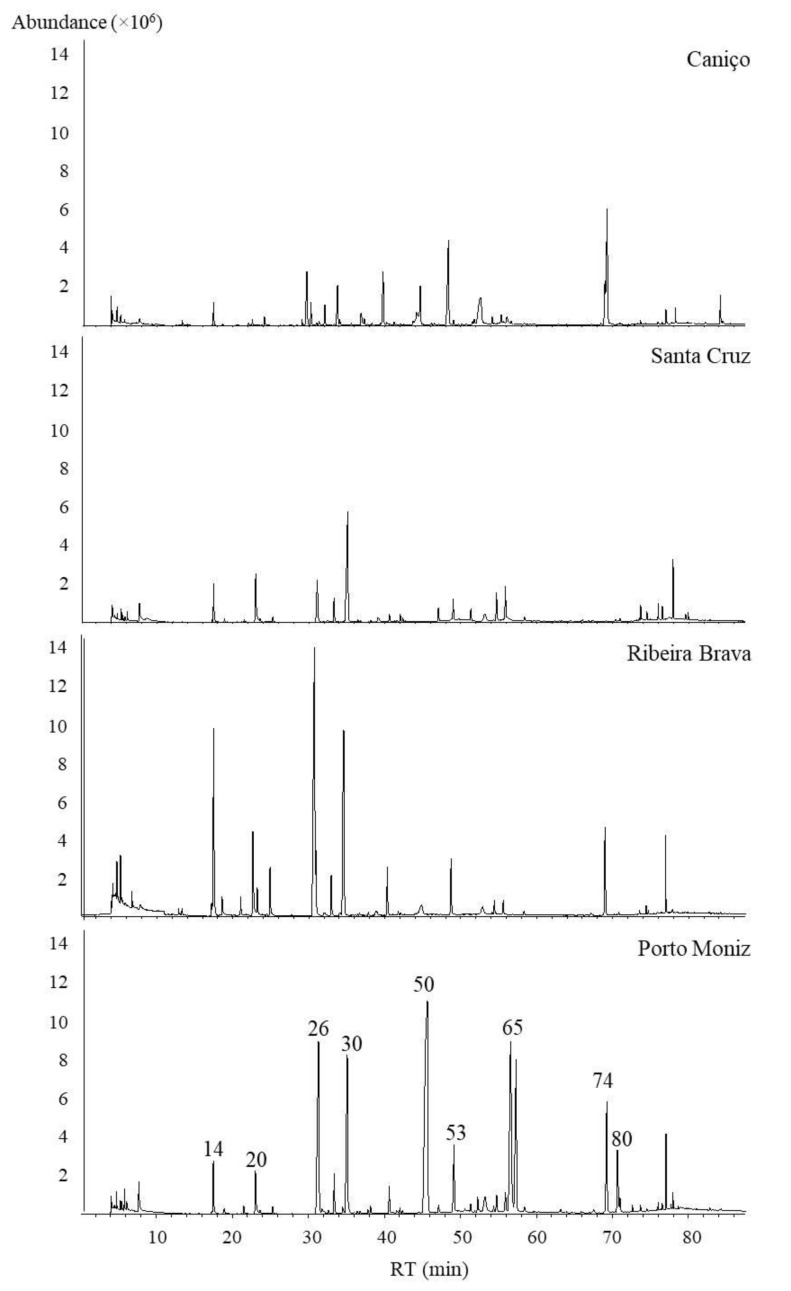
Total ion chromatograms obtained by HS-SPME/GC-qMS analysis of onions (*Allium cepa* L.) from Spanish variety from different geographical regions of Madeira Island. Attribution of the peak number is shown in Table 2.

**Figure 2 foods-09-00375-f002:**
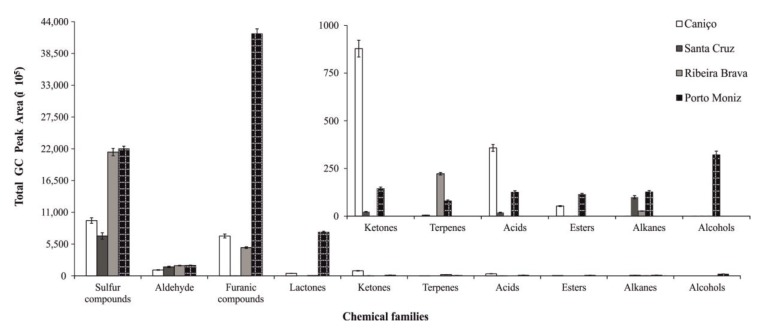
Volatile organic metabolites (VOMs) distribution by chemical groups identified in onions from Spanish cultivated at different geographic regions. Error bars represent mean standard deviation (*n* = 3).

**Figure 3 foods-09-00375-f003:**
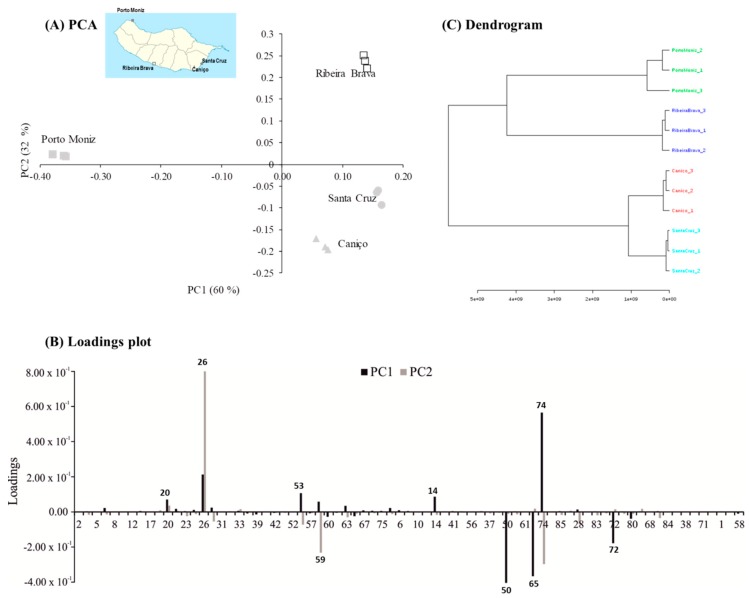
(**A**) Principal Component Analysis (PCA). First principal component (PC1) × second principal component (PC2) of scores scatter plot of *Allium cepa* L. according to geographical regions; (**B**) loading plot of the main source of variability between volatile profile and *Allium cepa* L. (attribution of the peak number shown in Table 2); (**C**) dendrogram of a cluster analysis in onions from four regions of Madeira Island (distance measure using Euclidean and clustering algorithm using ward).

**Figure 4 foods-09-00375-f004:**
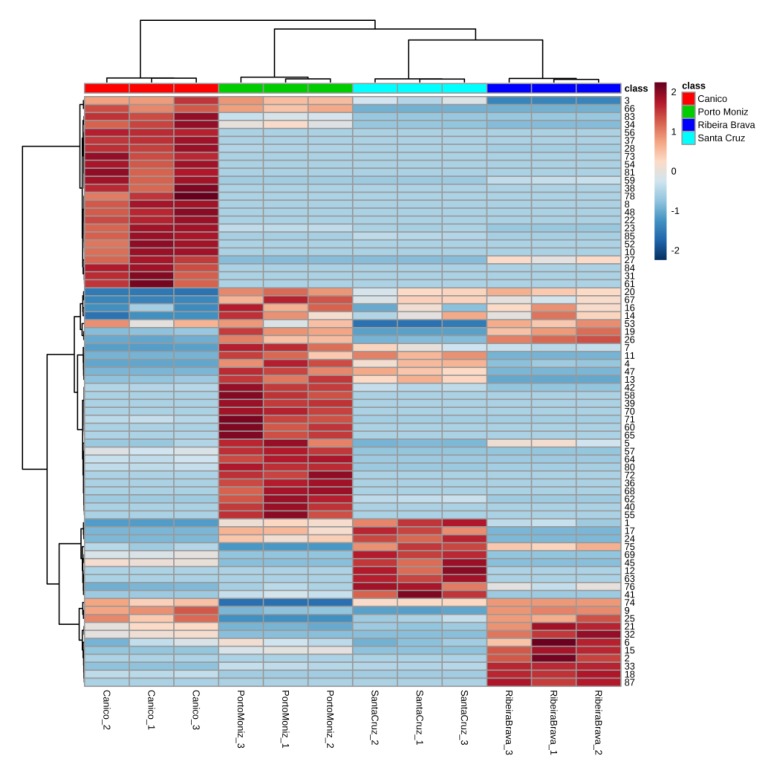
Clustering result in the form of a heatmap.

**Table 1 foods-09-00375-t001:** Geographic coordinates and climate conditions of the four regions of Madeira Island where the onions were cultivated.

Location	Latitude ^a^	Longitude ^a^	Altitude (m) ^a^	Mean Temperature (°C) ^b^	Mean Rainfall (mm) ^b^	Köppen-Geiger Climate Classification ^b^
Porto Moniz	32°52′0.06″ N	17°10′14.19″ W	40	18.9	559	Csa
Ribeira Brava	32°40′26.36″ N	17°3′3.57″ W	360	17.6	614	Csb
Caniço	32°39′6.14″ N	16°50′7.93″ W	163	17.6	614	Csb
Santa Cruz	32°41′21.41″ N	16°47′26.66″ W	31	18.8	592	Csa

Csa: hot-summer Mediterranean climate. Csb: warm-summer Mediterranean climate; ^a^ Information obtained from www.google.com/maps; ^b^ Information obtained from https://es.climate-data.org/.

**Table 2 foods-09-00375-t002:** The volatomic signature of *Allium cepa* L. from different geographic regions using headspace solid-phase microextraction (HS-SPME/GC-qMS).

Peak nº	RT (min) ^a^	KI calc ^b^	KI Lit ^c^	Chemical Groups/VOMs	Similarity (%)	GC Peak Area × 10^5^ (RSD)							
						Caniço		Santa Cruz		Ribeira Brava		Porto Moniz	
				**Sulfur Compounds**									
2	4.56	649	635	Methanethiol	90	-		-		14.66	(19)	-	
3	4.82	661	660	Carbon disulfide	83	129.23	(17)	59.71	(15)	-		107.84	(9)
5	5.82	802	770	1-Propanethiol	95	34.91	(17)	20.56	(12)	69.28	(17)	155.06	(17)
7	7.73	954	961	Thiophene	93	196.25	(4)	613.09	(17)	467.48	(5)	1154.56	(10)
8	12.93	1068	1068	Dimethyl disulfide	94	18.93	(13)	-		-		-	
11	16.40	1134	1127	3-Methyl-thiophene	86	-		4.41	(7)	-		4.70	(17)
12	16.95	1143	1150	Thiophane	86	-		13.03	(17)	-		-	
16	18.75	1173	1191	1-(Methylthio)ethanethiol	94	58.51	(16)	70.04	(19)	92.79	(9)	109.63	(11)
17	19.85	1190	1215	Methyl 2-propenyl disulfide	64	-		12.07	(14)	-		8.01	(14)
19	21.24	1213	1221	Methoxymethylbutanethiol	91	25.45	(16)	-		137.96	(17)	152.55	(19)
20	22.63	1235	1224	Methyl propyl disulfide	87	203.82	(12)	1280.73	(14)	1482.78	(9)	1844.03	(5)
21	23.20	1244	1228	Methyl isothiocyanate	82	201.68	(10)	83.93	(9)	380.48	(19)	48.61	(17)
23	24.64	1265	1253	2,4-Dimethyl-thiophene	74	228.34	(16)	13.99	(14)	-		31.06	(5)
24	25.10	1272	1273	Hexanethiol	64	-		222.73	(10)	-		115.76	(15)
26	30.84	1358	1370	Dipropyl disulfide	91	1528.80	(18)	2364.77	(13)	14,144.44	(7)	11201.60	(15)
27	31.10	1362	1364	Dimethyl trisulfide	97	812.81	(13)	-		338.43	(16)	-	
31	35.09	1423	1428	1-Propenyl propyl disulfide isomer	73	4.35	(11)	-		-		-	
32	36.05	1439	1429	Thenylthiol	91	29.44	(12)	-		75.95	(18)	-	
33	36.31	1444	1438	2-Pentylthiophene	74	-		27.18	(5)	215.22	(1)	34.47	(18)
36	36.81	1452	1458	Methional	79	-		-		-		291.57	(11)
39	38.28	1475	1476	2-Methyl-1,3-dithiane	78	-		-		-		480.56	(10)
40	38.71	1482	1506	Methyldihydrothiophenone	87	-		-		-		14.10	(9)
42	39.35	1492	1516	Methyl propyl trisulfide	78	35.29	(7)	45.86	(17)	-		313.73	(10)
47	41.98	1535	1520	Methyldithiofurane	78	-		95.76	(16)	-		155.88	(15)
52	46.52	1606	1606	Sulfurol	78	4.14	(13)	-		-		-	
53	48.90	1646	1622	Phenylethylthiol	95	3110.10	(17)	768.23	(10)	3310.32	(8)	2998.56	(19)
57	51.35	1686	1667	Methyl-(methyldithio)furan	80	74.19	(11)	-		-		302.68	(5)
59	52.50	1704	1700	3-(Acetylthio)-2-methylfuran	82	2629.08	(16)	-		354.53	(9)	44.83	(9)
60	53.19	1716	1723	Methionol	89	-		-		-		946.48	(19)
62	54.30	1734	1738	Dipropyl trisulfide	82	-		28.46	(16)	-		215.70	(13)
63	54.46	1737	-	3,5-Diethyl-1,2,4-trithiolane isomer	78	-		521.55	(10)	-		-	
64	54.71	1741	1759	3-Methylthiopropyl cyanide	78	133.36	(18)	-		-		902.24	(9)
67	58.41	1798	1785	Acetylthiophene	75	33.52	(10)	191.17	(15)	173.31	(12)	295.88	(18)
69	62.96	1896	1896	5-(Methylthio)-4-pentenenitrile	72	24.88	(12)	87.04	(5)	-		-	
75	70.43	2010	1998	Methyl methanethiosulfonate	75	19.28	(14)	65.46	(13)	44.65	(10)	-	
76	73.52	2029	2038	5-Methoxy-thiazole	72	40.10	(13)	314.49	(16)	124.92	(16)	82.21	(11)
				**Aldehyde**									
6	5.93	806	832	Butanal	72	105.00	(14)	93.41	(6)	177.39	(19)	115.90	(9)
9	13.36	1077	1084	Hexanal	72	122.25	(13)	6.62	(17)	119.68	(4)	22.95	(17)
10	14.02	1089	1101	2-Methyl-2-butenal isomer	84	28.82	(19)	-		-		-	
13	17.28	1149	1141	2-Methyl-4-pentenal isomer	93	8.74	(14)	33.48	(15)	-		55.06	(11)
14	17.42	1152	1155	2-Methyl-2-pentenal isomer	94	696.86	(11)	1284.04	(19)	1448.52	(16)	1618.87	(18)
25	27.57	1307	1280	Octanal	74	26.81	(17)	10.77	(18)	28.03	(15)	-	
41	39.12	1487	1484	Decanal	96	-		73.93	(19)	-		13.08	(15)
45	41.16	1521	1527	2-Nonenal isomer	94	16.10	(6)	39.40	(15)	-		-	
56	50.92	1680	1701	2,4-Nonadienal isomer	75	10.28	(2)	-		-		-	
				**Furanic Compounds**									
18	20.80	1205	1240	2-Pentylfuran	91	9.54	(14)	-		77.41	(5)	-	
37	37.25	1459	1458	Furfural	91	83.29	(8)	-		-		-	
48	43.73	1563	1560	5-Methyl-2-furfural	91	46.61	(16)	-		-		-	
50	44.59	1576	1580	Dimethylmethoxyfuranone	74	877.44	(15)	-		-		23,151.04	(19)
54	49.03	1649	1661	2-Furanmethanol	95	136.05	(14)	-		-		-	
61	54.09	1731	1710	5-Ethylfuran-2-(5H)-one	80	12.79	(18)	-		-		-	
65	56.53	1769	1745	Ethyl furanone	72	229.33	(15)	-		-		12,652.82	(17)
74	69.12	2002	2005	Acetyloxy-dimethylfuranone	72	5323.38	(10)	-		4723.21	(5)	6125.12	(2)
78	76.51	2045	2043	Furaneol	79	5.46	(19)	-		-		-	
85	84.15	2086	2087	5-Hydroxymethyl-2-furfural	83	167.64	(16)	12.16	(19)	-		-	
87	87.35	2102	2111	Methyl furaneol	86	-		-		85.87	(7)	-	
				**Ketones**									
28	32.12	1377	1388	2-Nonanone	86	684.65	(10)	22.86	(14)	-		28.48	(9)
55	50.57	1674	1645	Acetophenone	80	-		-		-		75.64	(14)
83	78.28	2055	-	DDMP	91	193.67	(12)	-		-		39.42	(17)
				**Lactones**									
66	57.27	1781	1770	γ-Heptalactone	98	84.42	(10)	-		-		71.65	(8)
72	67.41	1977	1977	Whiskey lactone isomer	78	47.51	(7)	-		-		6038.33	(12)
73	68.45	1995	1985	Whiskey lactone isomer	86	105.85	(13)	-		-		-	
80	77.02	2048	2042	γ-Nonalactone	86	198.88	(9)	-		52.20	(18)	1470.16	(4)
				**Terpenes**									
15	18.43	1168	1178	Limonene isomer	94	4.96	(6)	-		222.04	(11)	64.73	(9)
68	59.44	1820	1840	Geranyl acetone	86	-		-		-		14.95	(15)
				**Acids**									
34	36.59	1448	1449	Acetic acid	88	339.21	(16)	18.77	(17)	-		125.42	(17)
84	82.21	2076	2083	Octanoic acid	94	18.42	(10)	-		-		-	
				**Esters**									
22	24.54	1264	1288	Methyl heptanoate	78	18.15	(10)	-		-		-	
38	38.02	1471	1446	Propyl hexanoate	72	14.98	(18)	-		-		-	
70	63.19	1900	1915	3-Methylbutyl benzoate	74	-		-		-		74.31	(8)
71	63.96	1915	1906	Ethyl dihydrocinnamate	80	5.01	(18)	-		-		38.21	(18)
81	77.14	2049	2048	Methyl cinnamate	83	14.58	(15)	-		-		-	
				**Alkanes**									
1	4.47	644	-	3-Methyl-1,4-pentadiene	90	-		30.94	(15)	9.66	(19)	16.76	(7)
4	5.41	786	-	2,4-Hexadiene isomer	86	-		68.49	(17)	17.02	(18)	109.61	(16)
				**Alcohols**									
58	52.27	1701	1719	Undecanol	90	-		-		-		321.57	(16)
				**Unknown**									
29	33.16	1392	-	*m*/*z* = 148,106,41,45	-	619.12	(14)	1125.37	(15)	-		1044.64	(17)
30	34.74	1417	-	*m*/*z* = 148,106,41,45	-	1367.84	(1)	-		14,622.71	(9)	8517.34	(12)
43	39.75	1498	-	*m*/*z* = 43,41, 45, 47, 99, 154	-	2302.17	(15)	-		-		-	
44	40.51	1510	-	*m*/*z* = 154,112,43,47	-	-		844.13	(18)	-		455.38	(2)
46	41.42	1526	-	*m*/*z* = 117, 146, 45	-	-		33.75	(6)	-		27.30	(16)
49	44.36	1572	-	*m*/*z* = 87, 73, 45, 41	-	-		-		-		49.14	(8)
51	46.50	1605	-	*m*/*z* = 41,45, 73	-	19.75	(10)	-		-		-	
77	76.04	2043	-	*m*/*z* = 73,147,138	-	-		298.01	(19)	-		184.86	(12)
79	76.58	2046	-	*m*/*z* = 73,147,138	-	-		325.23	(16)	-		-	
82	77.94	2053	-	*m*/*z* = 113,112,111,97	-	-		-		-		409.26	(4)
86	84.34	2087	-	*m*/*z* = 134, 69	-	-		18.62	(13)	-		41.93	(19)

^a^ Retention time (RT) (min). RSD: relative standard deviation; VOMs: volatile organic metabolites. The results are expressed as mean values (*n* = 3; RSD < 20%). ^b^ Kovats index (KI) relative *n*-alkanes (C_8_ to C_20_) on a BP-20 capillary column. ^c^ Kovats index (KI) relative reported in literature for equivalent capillary column [18,20]. - Not detected; DDMP: 2,3-Dihydro-3,5-dihydroxy-6-methyl-4H-pyran-4-one.

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
