# Peer review of "Typicality Assessment of Onions (Allium cepa) from Different Geographical Regions Based on the Volatile Signature and Chemometric Tools"

_foods, 2020, doi:10.3390/foods9030375_

Round 1

Reviewer 1 Report

THE MANUSCRIPT IS WELL DESIGNED AND STRUCTURED AND EXPLORES AN IMPORTANT TOPIC FOR THE FIRST TIME, REFERENCE TO PREVIOUS STUDIES ARE PUNCTUAL AND COMPLETE.

THE ONLY DOUBT I HAVE IS ABOUT TABLE 2 AND HOW THE VOLATILE COMPOUNDS WERE IDENTIFIED. WHAT DOES THE COLUMN SIMILARITY MEAN?

IN LINES 128-130 IT IS STATED:  Single VOM peak was considered as  identified compound when its experimental spectrum corresponded with a score over 80% that present in the library; IT IS THE VALUE OF SIMILARITY?

BUT THERE ARE COMPOUNDS WITH LESS THAN 80%...PLEASE EXPLAIN

THE MANUSCRIPT NEEDS ALSO AN OVERALL ENGLISH REVISION.

 PLEASE REVISE ALSO THE LAYOUT OF TABLES.

Author Response

Reviewer #1, comment 1

“THE MANUSCRIPT IS WELL DESIGNED AND STRUCTURED AND EXPLORES AN IMPORTANT TOPIC FOR THE FIRST TIME, REFERENCE TO PREVIOUS STUDIES ARE PUNCTUAL AND COMPLETE.

THE ONLY DOUBT I HAVE IS ABOUT TABLE 2 AND HOW THE VOLATILE COMPOUNDS WERE IDENTIFIED. WHAT DOES THE COLUMN SIMILARITY MEAN? IN LINES 128-130 IT IS STATED: Single VOM peak was considered as identified compound when its experimental spectrum corresponded with a score over 80% that present in the library; IT IS THE VALUE OF SIMILARITY? BUT THERE ARE COMPOUNDS WITH LESS THAN 80%...PLEASE EXPLAIN”

Authors answer: The authors thank the reviewer's comment. The volatile organic metabolites (VOMs) were identified using standards when available; by comparison of VOM mass spectra with the data system library (NIST, 2005 software, Mass Spectral Search Program v.2.0d; Nist 2005, Washington, DC) and by Kovats index (KI). KI was calculated using C8–C20 n-alkanes series and the values were compared, when available, with values reported in the scientific literature for similar/equivalent columns (DB-Wax, HP-20M, Supelcowax 10, CB-Wax, Stabilwax, Carbowax, HP-Innowax, Rtx-WAX, PE-WAX, RH-WAX, ZB-WAX, TRWAX).

The “similarity” represents the percentage of “equality” between the mass spectrum of the compound in the analysed sample compared to the mass spectrum of the data system (NIST). A percentage greater than 80% was established to consider the compound as identified. However, to confirm this identification, pure standards were analysed when available in addition to the determination of Kovats indices (KI) comparing them with those reported in the literature. For similarity percentages below 80%, these two parameters were used in addition to the published data for the volatile composition of onions. In these cases, a tentative identification based on comparative data and on the interpretation of the mass spectrum of each compound was carried out.

Reviewer #1, comment 2

“THE MANUSCRIPT NEEDS ALSO AN OVERALL ENGLISH REVISION.

 PLEASE REVISE ALSO THE LAYOUT OF TABLES.”

Authors answer: As suggested by the reviewer, the English revision has been performed and the layout of the tables has been checked.

Reviewer 2 Report

This is a well-written manuscript describing a method for establishing the “volatile signature” of onions from different locations. It can be useful for authentication of onions coming from given geographical regions. The novelty of the described research work is not high because the Authors use standard analytical procedures, however it is worth publishing due to its practical usefulness. It can interesting also for the researchers studying dependences between the cultivating conditions and the secondary metabolites composition of different plants.

I didn’t find any errors in the manuscript. All parts are written correctly. The description of experiments is clear and contains enough details to enable their reproducing in another laboratories.

I have only one question concerning the sample handling. Did I understood correctly that fresh onions were frozen to -20 oC before slicing and extraction? According to my knowledge, deep freezing can change the composition of the secondary metabolites. Onions are usually stored at room temperature or under mild refrigeration.

After clarifying this point the manuscript will be suitable for publication.

Author Response

Reviewer #2, comment 1

This is a well-written manuscript describing a method for establishing the “volatile signature” of onions from different locations. It can be useful for authentication of onions coming from given geographical regions. The novelty of the described research work is not high because the Authors use standard analytical procedures, however it is worth publishing due to its practical usefulness. It can interesting also for the researchers studying dependences between the cultivating conditions and the secondary metabolites composition of different plants.

I didn’t find any errors in the manuscript. All parts are written correctly. The description of experiments is clear and contains enough details to enable their reproducing in another laboratories.

I have only one question concerning the sample handling. Did I understood correctly that fresh onions were frozen to -20 °C before slicing and extraction? According to my knowledge, deep freezing can change the composition of the secondary metabolites. Onions are usually stored at room temperature or under mild refrigeration.

After clarifying this point the manuscript will be suitable for publication.

Authors answer: Regarding the comment of Reviewer, a previous report has investigated the influence of low storage temperatures (+4 °C and -20 °C and storage at room temperature (25 °C) on volatile profiles of different varieties of virgin olive oils (VOO) (Bubola, et al., 2014). These authors found that the VOO volatile profile showed quite good stability during storage time (1 year) at all investigated temperatures and the total volatile compounds (aldehydes, alcohols, and esters) in almost all stored samples were unchanged compared to fresh oils. The authors concluded that cold and frozen storage of VOO samples may be recommended since changes in volatile compounds profile are less at these temperatures in comparison with room temperature.

In addition, in the current study, all onion samples were processed and stored in the same conditions, so the obtained results were not biased due to the temperature of storing.

Bubola, K. B., Koprivnjak, O., Sladonja, B., & Belobrajić, I. (2014). Influence of storage temperature on quality parameters, phenols and volatile compounds of Croatian virgin olive oils. Grasas y aceites, 65(3), e034-1.
